# Gustatory-mediated avoidance of bacterial lipopolysaccharides via TRPA1 activation in *Drosophila*

**Alessia Soldano[1,2,3], Yeranddy A Alpizar[1], Brett Boonen[1], Luis Franco[2,3,4], Alejandro López-Requena[1], Guangda Liu[2,3], Natalia Mora[2,3], Emre Yaksi[4,5], Thomas Voets[1], Rudi Vennekens[1], Bassem A Hassan[2,3,6,7]\*, Karel Talavera[1]\***

[1]Laboratory of Ion Channel Research and TRP Research Platform Leuven, Department of Cellular and Molecular Medicine, KU Leuven, Leuven, Belgium; [2]VIB Center for the Biology of Disease, VIB, Leuven, Belgium; [3]Center for Human Genetics, University of Leuven School of Medicine, Leuven, Belgium; [4]Neuroelectronics Research Flanders, Leuven, Belgium; [5]Kavli Institute for Systems Neuroscience and Centre for Neural Computation, NTNU, Trondheim, Norway; [6]Institut du Cerveau et de la Moelle Epinière, Hôpital Pitié-Salpétrière, Paris, France; [7]Ecole Doctorale Cerveau Cognition Comportement, Université Pierre et Marie Curie, Sorbonne Universités, Paris, France

\*For correspondence: bassem. hassan@icm-institute.org (BAH); karel.talavera@med.kuleuven.be (KT)

**Competing interests:** The authors declare that no competing interests exist.

**Abstract** Detecting pathogens and mounting immune responses upon infection is crucial for animal health. However, these responses come at a high metabolic price (*McKean and Lazzaro, 2011*, *Kominsky et al., 2010*), and avoiding pathogens before infection may be advantageous. The bacterial endotoxins lipopolysaccharides (LPS) are important immune system infection cues (*Abbas et al., 2014*), but it remains unknown whether animals possess sensory mechanisms to detect them prior to infection. Here we show that *Drosophila melanogaster* display strong aversive responses to LPS and that gustatory neurons expressing Gr66a bitter receptors mediate avoidance of LPS in feeding and egg laying assays. We found the expression of the chemosensory cation channel dTRPA1 in these cells to be necessary and sufficient for LPS avoidance. Furthermore, LPS stimulates *Drosophila* neurons in a TRPA1-dependent manner and activates exogenous dTRPA1 channels in human cells. Our findings demonstrate that flies detect bacterial endotoxins via a gustatory pathway through TRPA1 activation as conserved molecular mechanism.

## Results and discussion

In the past decade, increasing attention has been paid to the interactions between the immune and nervous systems (*McMahon et al., 2015*). In particular, there is evidence that sensory neurons can directly detect bacterial components as potentially damaging stimuli, and initiate acute inflammatory and nocifensive responses (*Chiu et al., 2013*; *Meseguer et al., 2014*). It has been shown that the Gram-negative bacterial wall component LPS induces hygienic grooming in *Drosophila*, an important behavioral defense against pathogens, via contact chemosensation (*Yanagawa et al., 2014*). Thus, LPS may represent important sensory cues of food contamination with Gram-negative bacteria. To test whether LPS can be perceived by flies during food ingestion we used a binary food choice assay (*Isono and Morita, 2010*) (*Figure 1—figure supplement 1A*). We found that control flies displayed significant avoidance towards food supplemented with LPS (*Figure 1A* and *Figure 1—figure*

**eLife digest** An immune system can fight bacterial infections, ensuring an animal's health and survival. However, mounting an immune response to a bacterial infection requires a lot of energy. It also can be potentially dangerous if the immune system becomes too active. Therefore, avoiding bacteria and not getting infected to begin with may be a better strategy to stay healthy. Fruit flies, like humans, can detect dangerous substances in the environment via their sense of smell, but it is not known whether they also detect disease-causing organisms through their sense of taste.

Bacterial molecules called lipopolysaccharides (LPS) can alert the immune system to the presence of dangerous bacteria. Previous studies have found that when flies get in contact with LPS they begin cleaning themselves, which might help prevent infection. However it was not clear how the flies actually detected the LPS. Now, Soldano et al. show that fruit flies can taste LPS and avoid eating or laying eggs on food contaminated with LPS and bacteria. A series of experiments showed that when a fly tastes LPS it stimulates bitter-sensing neurons in the fly's mouth and throat. The experiments also revealed that the protein that activates these neurons in response to LPS is the same protein that acts in humans as detector of pungent chemicals contained in ordinary food items like mustard, garlic and wasabi. This suggests this protein, called TRPA1, is part of a key survival mechanism that has been preserved in many species throughout evolution.

Soldano et al. showed that a fly's senses and nervous system are actively involved in protecting it from bacterial infection. This is particularly important to flies, because unlike humans they don't develop resistance to future infections with the same bacteria. Future studies are needed to determine if flies use their sense of taste to detect other chemicals that are signs of infections. Additionally, studies are needed to determine if the activated bitter-sensing nerves alert the fly's immune system to a potential infection.

*supplement 1B*). Because LPS is non-volatile we determined if this avoidance is mediated by gustatory neurons known to detect aversive compounds (Gr66a) (*Marella et al., 2006*). Blocking neurotransmission in these neurons by expressing the light chain of tetanus toxin (TNT) abolished avoidance of LPS (*Figure 1B*), indicating that flies can detect LPS through a gustatory mechanism.

A subset of Gr66a neurons innervating the labral sense organ and the labellum express TRPA1 (*Kim et al., 2010*; *Kang et al., 2011*), a chemosensory cation channel (*Story et al., 2003*; *Nilius et al., 2012*; *Zygmunt and Högestätt, 2014*) that mediates acute nocifensive responses to LPS in mice (*Meseguer et al., 2014*) and avoidance of bitter and noxious compounds in *Drosophila* (*Kim et al., 2010*; *Kang et al., 2010*; *Du et al., 2015*). We tested whether TRPA1 mediates gustatory avoidance of LPS in flies. We found that loss of *dTrpA1* ($w^{1118}$;*dTrpA1$^1$*, *Figure 1C* and *dTrpA1$^1$/dTrpA1$^{ins}$*, *Figure 1—figure supplement 1C*) and pan-neuronal *dTrpA1* knockdown by two independent RNAi lines (*Figure 1—figure supplement 1D*) lead to impaired avoidance of LPS. Therefore, neuronal expression of *dTrpA1* is required for LPS avoidance. Furthermore, *Gr66a*-specific knockdown of *dTrpA1* abolished the LPS-induced behavior (*Figure 1D*), and restoration of *dTrpA1* expression using either of two different *dTrpA1* isoforms (A and B) in the entire *dTrpA1* pattern or only in Gr66a gustatory neurons of *dTrpA1$^1$/dTrpA1$^{ins}$* flies rescued the avoidance of LPS (*Figure 1E*).

Female flies use gustatory detection of non-volatile compounds via Gr66a neurons to select oviposition sites (*Joseph and Heberlein, 2012*). In a binary oviposition choice assay control females showed preference for control food over food supplemented with LPS (*Figure 2A*). This behavior was lost in *dTrpA1$^{-/-}$* flies (*Figure 2A*), upon silencing *Gr66a*-expressing neurons (*Figure 2B*), and in *Gr66a*-specific *dTrpA1* knockdown flies (*Figure 2C*). Altogether, these data show that LPS is avoided during feeding and oviposition and that *dTrpA1* expression in bitter-sensing gustatory neurons is necessary and sufficient for LPS avoidance. These data indicate that a TRPA1-dependent mechanism of avoidance of LPS may serve flies to prevent infection with Gram-negative bacteria. Indeed, we found that control animals, but not *dTrpA1$^{-/-}$* mutants, preferred laying eggs on control food, rather than on food contaminated with *E. coli* (*Figure 2D*).

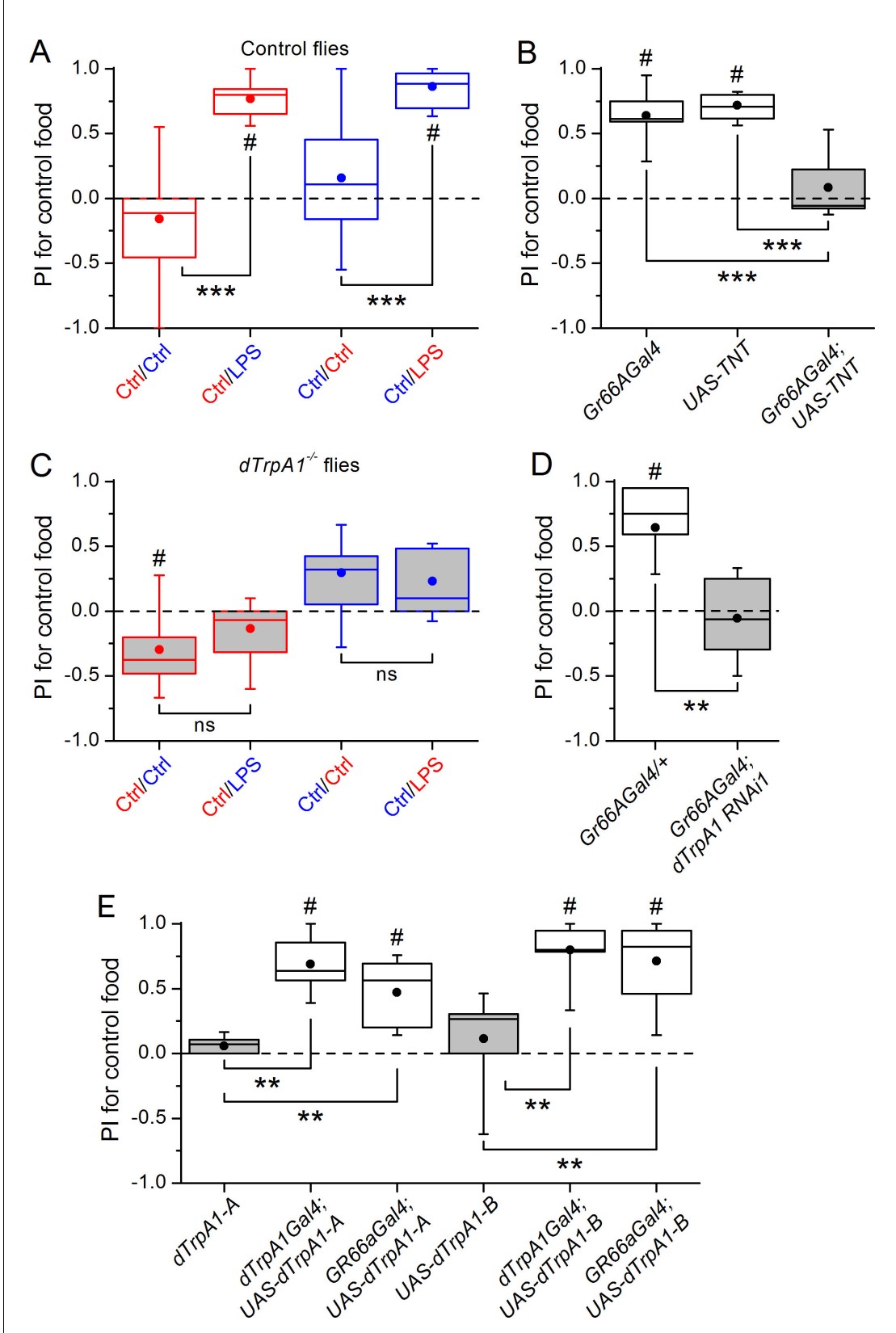

**Figure 1.** Gustatory *dTrpA1*-dependent avoidance of LPS in *Drosophila melanogaster*. (A,C) Preference index (PI) of control *CS10w^1118* (A) and *dTrpA1*-deficient (C) male flies for control food over LPS-supplemented food. PI was calculated for the consumption of the control solution mixed with the color

*Figure 1 continued on next page*

*Figure 1 continued*

of the corresponding data symbol (n ≥ 6). (B) PI for control food of *Gr66a>TNT* flies and in the corresponding driver-only and responder-only control flies (n ≥ 6). (D) PI for control food of *Gr66a>dTrpA1 RNAi* flies (n ≥ 4–8). (E) Rescue of LPS avoidance in *dTrpA1/dTrpA1^ins^; dTrpA1>dTrpA1* and *dTrpA1^1^/dTrpA1^ins^; Gr66a>dTrpA1-A/B* flies (n ≥ 5). *P < 0.05; **P < 0.01; ***P < 0.001; 'ns', P > 0.05 (two-tailed Mann-Whitney *U* test). #, statistically significant differences from the no-preference zero level (two-tailed *t* test).

The following figure supplement is available for figure 1:

**Figure supplement 1.** *dTrpA1*-dependent aversion to LPS in *Drosophila.*

dTRPA1 is expressed in the mouthpart in a subset of gustatory neurons in the esophagus (*Figure 3A,B* and *Kang et al., 2010*) and in a few neurons in the labellum that also express Gr66a (data not shown and *Kang et al., 2011*). However, no co-localization between *dTrpA1* and *Gr66a* was observed in the leg (*Figure 3—figure supplement 1*). To test whether avoidance of LPS is mediated by labellar or esophageal chemosensors we performed a proboscis extension reflex (PER) assay in wild type animals. LPS did not inhibit PER (*Figure 3—figure supplement 2*), suggesting that avoidance of LPS is not mediated by the labellar neurons, but by the gustatory neurons of the esophagus. We attempted to record neuronal responses in these neurons using flies expressing the genetically encoded $Ca^{2+}$ indicator GCaMP6m in *Gr66a* neurons but $Ca^{2+}$ imaging access to these neurons proved impossible in our assays. Next, we attempted direct brain stimulation. All preparations (5/5) responded robustly to the application of the dTRPA1 agonist allyl isothiocyanate (AITC) or the classical bitter compound caffeine, indicating that they were healthy. However, application of LPS gave varying results (small responses in 40% (2/5) of the flies; data not shown) precluding definitive conclusions. Therefore, in order to further test whether dTRPA1 mediates responses to LPS in vivo we monitored intracellular $Ca^{2+}$ dynamics in the ventral nerve cord of larvae, a preparation that allows better accessibility of chemical stimuli. Application of LPS or the dTRPA1 agonist allyl isothiocyanate (AITC) induced robust $Ca^{2+}$ responses, effects that were strongly reduced by incubation with the TRPA1 inhibitor HC030031 (*Figure 4—figure supplement 1*). In contrast, HC030031 did not affect the responses to a depolarizing solution containing high $K^+$ concentration.

To test the role of TRPA1 in neuronal responses to LPS at the cellular level we examined primary cultures of larva brain neurons (*Figure 4—figure supplement 2*, *Harzer et al., 2013*) expressing the $Ca^{2+}$ indicator GCaMP5 under the control of *dTrpA1Gal4*. LPS (30 µg/ml) reversibly stimulated more than 40% of neurons isolated from control larvae (122/289), and 80% of these were also activated by the dTRPA1 agonist N-ethyl maleimide (NEM, 300 µM) (*Kim and Cavanaugh, 2007*) (*Figure 4A,B*). Notably, the proportion of neurons responding to both agents was strongly reduced in cultures derived from *dTrpA1^1^* null animals (20/153, P < $10^{-4}$, Fisher exact test), as well as by pharmacological inhibition of dTRPA1 with HC030031 in neurons isolated from control animals (4/32, P < $10^{-3}$, Fisher exact test). Intriguingly, the responses to LPS or NEM were not fully abolished by genetic or pharmacological ablation of dTRPA1. This suggests that dTRPA1 mediates some, though not all, $Ca^{2+}$ responses of larva brain neurons to these compounds. To verify this indication in other experimental settings we evaluated the responses of cells isolated from brains of wild type larvae using the ratiometric $Ca^{2+}$ indicator Fura2, and AITC as reference TRPA1 agonist. Application of LPS (60 µg/ml) reversibly stimulated more than 70% (39/54) of neurons isolated from control larvae (*Figure 4—figure supplement 3A,E,F*). A large proportion of LPS-sensitive neurons (23/39, 59%) were also activated by 100 µM AITC (*Figure 4—figure supplement 3A,E*), which indicates that LPS stimulates cells functionally expressing dTRPA1 channels. No $Ca^{2+}$ response was observed when LPS was applied in the absence of extracellular $Ca^{2+}$ (0/14, P < $10^{-4}$, Fisher exact test; *Figure 4—figure supplement 3B,E,F*). This indicates that LPS-induced responses result from $Ca^{2+}$ influx through channels in the plasma membrane, rather than from release from intracellular stores. To directly assess whether LPS induces neuronal responses through the activation of dTRPA1, we tested its effects on wild type neurons in the presence of HC030031 (*Figure 4—figure supplement 3C*). In these experiments, LPS-induced responses were significantly less frequent (9/32, 28%, P < $10^{-3}$, Fisher exact test, *Figure 4—figure supplement 3E*) and smaller in amplitude (P < $10^{-3}$, Mann-Whitney U test, *Figure 4—figure supplement 3F*) than in control experiments. To confirm these data genetically, we tested the effects of LPS on neurons isolated from *dTrpA1*-null larvae. These cells responded to

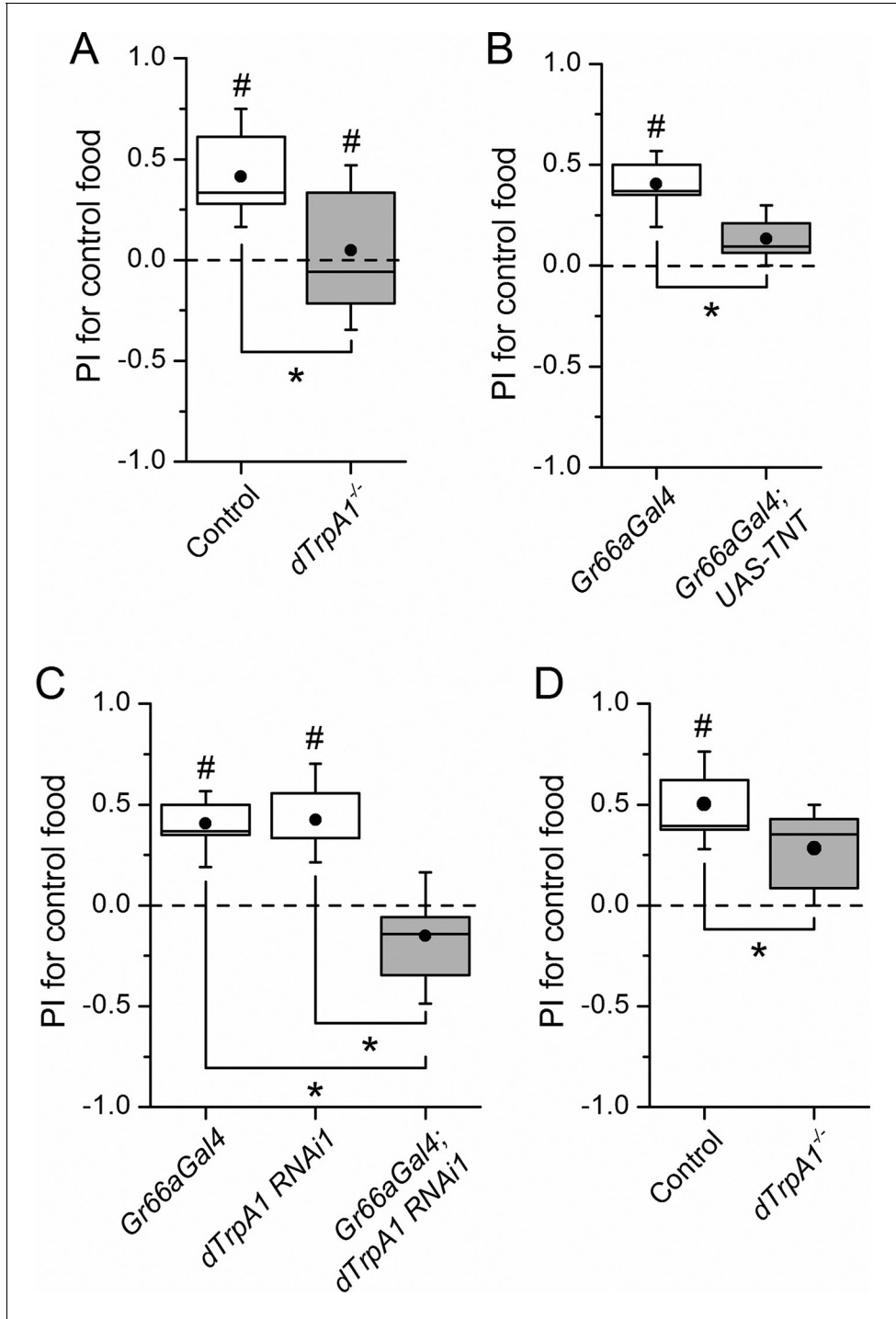

**Figure 2.** *dTrpA1* expression in gustatory neurons is required for avoidance of LPS during oviposition. (A) Preference index for oviposition in control food of wild type versus *dTrpA1*[-/-] flies (n ≥ 5). (B) Oviposition preference of *Gr66a>TNT* flies (n ≥ 6). (C) Oviposition preference of *Gr66a>dTrpA1 RNAi* flies (n ≥ 6). (D) Oviposition preference of wild type versus *dTrpA1*[-/-] flies in presence of *E. coli*. (n ≥ 8) *P < 0.05; **P < 0.01; ***P < 0.001 (two-tailed Mann-Whitney *U* test). #, statistically significant differences from the no-preference zero level (two-tailed *t* test).

LPS with significantly lower frequency (5/23, 22%, P < 10[−4], Fisher exact test, *Figure 4—figure supplement 3D,E*) and amplitude (P < 10[−4], Mann-Whitney U test, *Figure 4—figure supplement 3F*) than control neurons. Importantly, the subpopulation of neurons responsive to both LPS and AITC in cultures from control larvae was absent in cultures from *dTrpA1*-null animals (*Figure 4—figure*

*supplement 3E*). Taken together, these data indicate that dTRPA1 channels expressed in the plasma membrane mediate at least part of the $Ca^{2+}$ influx triggered by stimulation with in *Drosophila* neurons. Interestingly, as for the experiments with NEM, neither the LPS- nor the AITC-induced responses were fully absent in *TrpA1*-null neurons. This demonstrates that receptors other than TRPA1 may be also sensitive to these compounds in cultured brain neurons. This is in line with previous reports showing that sensory neurons isolated from *Trpa1* knockout mice showed reduced, but not completely abrogated responses to LPS (*Meseguer et al., 2014*), and that TRPA1 is not the only target of electrophilic compounds in sensory neurons (*Alpizar et al., 2014*; *Everaerts et al., 2011*; *Gees et al., 2013*; *Ohta et al., 2007*; *Salazar et al., 2008*). Here it is important to note that we used the larval fillet preparation and the neuron cultures as experimental models for native functional expression of dTRPA1.

Finally, we determined whether dTRPA1 can be activated by LPS in the HEK293T heterologous expression system. We found that 60 µg/ml LPS induced only very few responses in non-transfected cells (7/110), but stimulated a significantly larger fraction of cells transiently transfected with dTRPA1-A (42/74, P < $10^{-4}$, Fisher exact test) or dTRPA1-B (14/67, P = 0.007, Fisher exact test) (*Figure 4—figure supplement 4*). These responses were more variable in amplitude and less frequent than those triggered by AITC, indicating that LPS is a relatively weak agonist of dTRPA1 channels (*Alpizar et al., 2013*). Further evidence for dTRPA1 activation by LPS was obtained in whole-cell patch-clamp experiments, in which application of LPS significantly enhanced both outward and inward currents in dTRPA1-A transfected HEK293T cells, but not control cells (*Figure 4C,D*). Application of HC030031 reduced the amplitude of currents recorded in the presence of LPS (35 ± 4% at -75 mV, n = 6, *Figure 4C*), further confirming the TRPA1-dependence of these responses. Previous results suggest that LPS activates mouse TRPA1 channels by inducing mechanical perturbations in the plasma membrane upon insertion of the lipophilic moiety of the molecule (*Meseguer et al., 2014*). It is conceivable that LPS activates dTRPA1 channels via the same mechanism, but additional experiments are required to verify this.

Taken together, our data demonstrate that fruit flies possess a gustatory mechanism underlying the detection and avoidance of LPS. The avoidance of LPS-contaminated food during feeding and oviposition may serve to prevent Gram-negative bacterial infections, potentially compensating for the lack of adaptive immunity in these animals. The fact that this sensory mechanism exploits dTRPA1 suggests a broadly conserved principle whereby these channels play a crucial role in LPS detection by sensory neurons in flies and mammals, regardless of the particular sensory modality involved. Our findings, together with previous evidence of an olfactory-based detection of secondary metabolites of Gram-positive bacteria (*Stensmyr et al., 2012*), underscore the need to consider the function of the sensory nervous system as a crucial part of a broad mechanistic understanding of pathogen-host interactions.

## Materials and methods

### *Drosophila* stocks

*Drosophila melanogaster* strains were raised on standard cornmeal/agar medium supplemented with dry yeast at 25 °C with a 12 hr light/dark cycle. The wild type stock was a $w^{1118}$ strain. The following stocks were obtained from Bloomington Stock Center: $w^{1118}$;*dTrpA1*[1] (BL26504), *dTrpA1 RNAi1* (BL31384), *w\*;dTrpA1*[1], *dTrpA1-Gal4* (BL36922), *w\*; dTrpA1-Gal4* (BL 27593), UAS-RFP (BL 27391), *UAS-GCaMP5* (BL 42037), *UAS-GCaMP6m* (BL 42748). The Gr66a-IRES-GFP vector was kindly provided by Kristin Scott, the *NSyB-Gal4;UAS-GCaMP3* was obtained by Patrick Verstreken. The lines *Gr66aGal4*, *UAS-dTrpA1(A);dTrpA1*[ins],*dTrpA1GAL4*, *UAS-dTrpA1(B);dTrpA1*[ins],*dTrpA1-Gal4* and the *dTrpA1 RNAi 2* were kindly provided by the laboratory of Paul Garrity.

### Binary choice food preference

Groups of 20–30 adult males (2–7 days old) were starved for 20 hr in plastic tubes provided with humidified filter paper. After starvation, the animals were allowed to feed on a microtiter dish containing wells alternating 100 mM sucrose alone or with 1 mg/ml LPS, mixed with either a red or blue dye (Supplementary *Figure 1*, left). This concentration of LPS was chosen in accordance with a previous study reporting concentrations between 250 and 800 µg/ml in water used to rinse fruits and

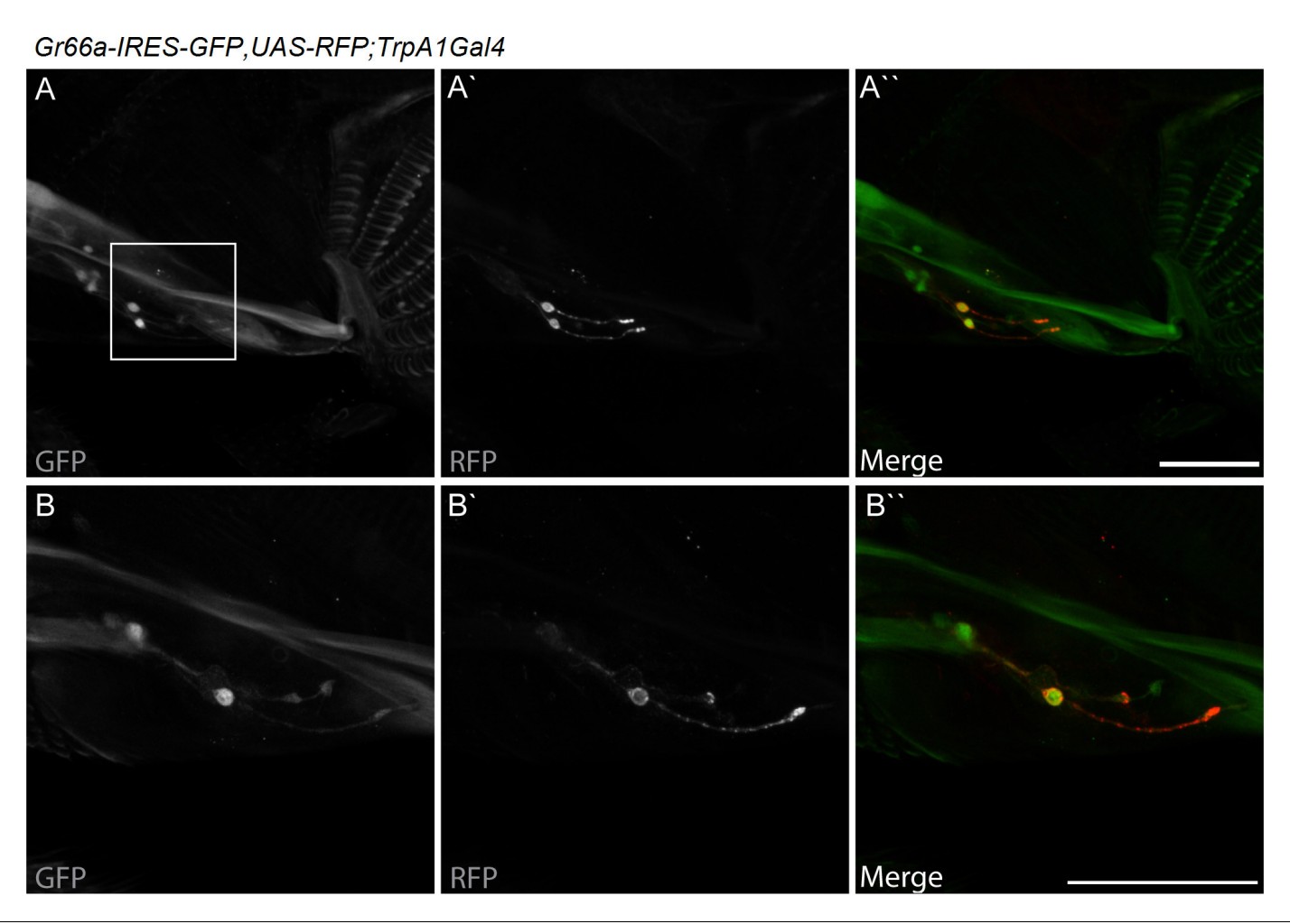

*Gr66a-IRES-GFP,UAS-RFP;TrpA1Gal4*

**Figure 3.** *dTrpA1* is expressed in a subset of *Gr66a*-expressing neurons that can be directly stimulated by LPS. (**A**) Immunofluorescence analysis of *Gr66a-IRES-GFP,UAS-RFP;dTrpA1Gal4* adult proboscis. Anti-GFP immunohistochemistry (in green in the Merge panel) labels taste neurons while anti-RFP (in red in the Merge panel) labels *dTrpA1*-expressing cells. Scale bar = 50 μm. (**B**) High magnification image of esophageal neurons expressing *Gr66a* and *TrpA1*.

The following figure supplements are available for figure 3:

**Figure supplement 1.** dTRPA1 is not expressed in tarsal Gr66a-expressing neurons.

**Figure supplement 2.** LPS does not alter the proboscis extension reflex.

vegetables contaminated with *E. coli* (*Wang et al., 2011*). The concentrations in the surface of the food would be therefore much higher than these values, and hence the concentration of 1 mg/ml is likely to be relevant for real scenarios. The feeding preference was assessed by examining the colors of the abdomen and by classifying the flies as red, blue or purple (Supplementary *Figure 1*, right). The preference index (PI) for the control-containing solutions was calculated as: $(n_{Control} - n_{Test})/(n_{Control} + n_{Test})$, where $n_{Control}$ and $n_{Test}$ are the number of flies that ate the control solution (in red or blue color) and the test solution (containing control or LPS, both in red or blue color), respectively. The test solutions were colored with food dyes that showed comparable results: red (food dye or Sulforhodamine B sodium salt, 1 mg/ml) and blue (food dye or Erioglaucine disodium salt, 0.16 mg/ml). Preference data was always represented in box charts in which lines and the dots inside the boxes represent the median and the mean of the data, respectively.

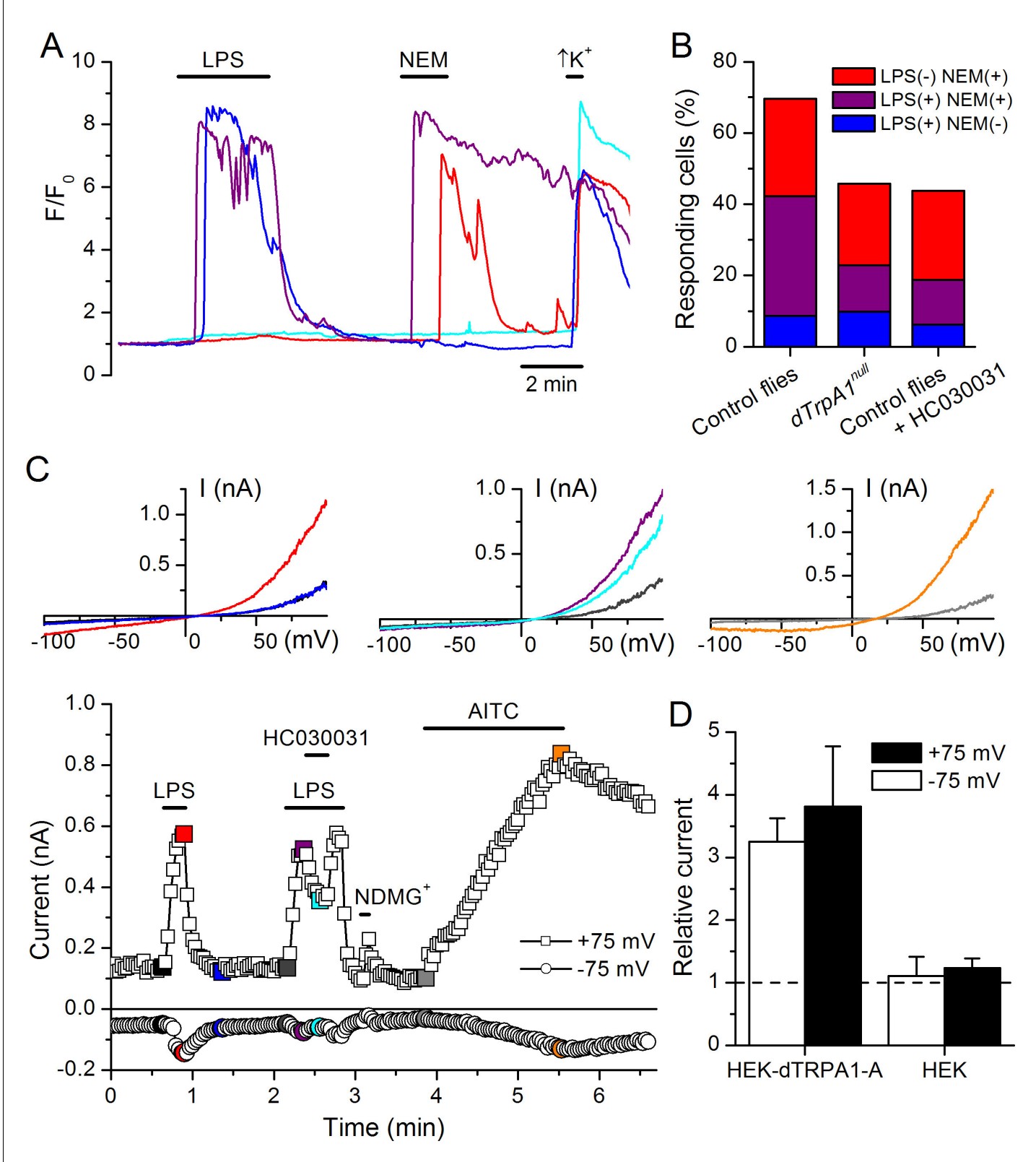

**Figure 4.** LPS stimulates dTRPA1. (**A**) Intracellular $Ca^{2+}$ imaging in cultured brain L3 neurons expressing RFP and GCaMP5 under the control of *dTrpA1Gal4* in baseline conditions and during extracellular perfusion with LPS (30 μg/ml), NEM (300 μM) or high $K^+$ (50 mM KCl). (**B**) Proportions of neurons responding to LPS (red), NEM (blue) or both (purple). (**C**) Time course of the currents amplitude measured at +75 and -75 mV during whole-cell patch-clamp recording in dTRPA1-A transfected HEK293T cell. The colored data points correspond to current traces shown at the top of the panel. (**D**)
*Figure 4 continued on next page*

*Figure 4 continued*

Average effects of LPS on the current amplitudes recorded at +75 and -75 mV in dTRPA1-A-transfected (n = 7) and non-transfected (n = 3) HEK293T cells. For each cell, the current amplitudes measured in the presence of LPS were normalized to those measured in control condition.

The following figure supplements are available for figure 4:

**Figure supplement 1.** LPS stimulates dTRPA1 in vivo.

**Figure supplement 2.** Images of a culture of cells from third instar larvae brains.

**Figure supplement 3.** LPS stimulates Drosophila neurons in vitro in a dTRPA1-dependent manner.

**Figure supplement 4.** LPS-induced responses in HEK293T cells transfected with dTRPA1 channels.

## Egg-laying assay

Adult flies were allowed to mate for 48 hr prior to test. Males were subsequently discarded and females were transferred to a test tube on top of a 35 mm dish containing food. The food was covered on one side with control solution (water) and on the other side with test solution. The test solution consisted of either *E. coli* or 1 mg/ml LPS in water. The flies were allowed to lay eggs for 20 hr at 25 °C in a light-proof chamber, after which the amount of eggs on each side of the plate was counted. A preference index (PI) for laying eggs in the control side was calculated as: $(n_{ControlSide} - n_{OtherSide})/(n_{ControlSide} + n_{OtherSide})$, where $n_{Control}$ and $n_{OtherSide}$ are the number of eggs laid in the control and in the other side (containing control, *E. coli* or LPS), respectively.

## Proboscis extension reflex assay

Two to seven day old flies were starved overnight on wet wipes, anaesthetized on ice, and gently pushed into a pipette tip. The flies were positioned with the head and the proboscis protruding outside the tip edge and the body immobilized within the tip. Flies recovered for 5–10 min before the test and then satiated with water. The solutions were presented as a liquid drop on a syringe tip and the proboscis was touched with the drop for 5 consequent times. During the assay flies were first stimulated with water, to exclude that any response would be due to thirst, and subsequently with 100 mM sucrose alone, as positive control, 100 mM sucrose + 1 mg/ml LPS and finally with 100 mM sucrose again. The extension frequency was calculated as mean frequency of all the samples (n = 19 animals).

## *In vivo* intracellular $Ca^{2+}$ imaging on the *Drosophila* sub-esophageal ganglion

For each experiment, a fly was secured to an aluminum chamber with wax. The antennae and the surrounding cuticle were removed. The brain was immerged into a bath containing adult hemolymph-like (AHL) solution. The proboscis was partially severed to allow easier penetration of the chemicals. The sub-esophageal ganglion was imaged in vivo from the dorsal side while constantly perfusing the brain with oxygenated AHL saline. The saline containing (in mM): 108 NaCl, 5 KCl, 2 $CaCl_2$, 8.2 $MgCl_2$, 4 $NaHCO_3$, 1 $NaH_2PO_4$, 5 trehalose, 15 ribose, 5 HEPES (pH 7.3). GCaMP fluorescence was imaged at 4 Hz using an EMCCD camera (Hamamatsu Photonics) installed on an Olympus BX51 fluorescence microscope (Olympus Corporation).

## *Drosophila* larval fillet preparation and *in vivo* intracellular $Ca^{2+}$ imaging

In vivo intracellular $Ca^{2+}$ imaging was performed in larvae fillets, consisting on surgically exposed brains and ventral nerve cords of wandering third instar larvae. Briefly, larvae were immobilized using pins and stretched out lengthwise in a dish coated with silica and filled with a HL3 solution containing (in mM): 70 NaCl, 5 KCl, 20 $MgCl_2$, 10 $NaHCO_3$, 5 trehalose, 115 sucrose and 5 HEPES (pH 7.3). The larval skin was cut vertically along the dorsal midline towards the rostral end of the larva using spring scissors. At the rostrum of the animal horizontal incisions were made to the left and right. This created a left and a right flap in the body wall. These flaps were then pulled and pinned on the

side of the larva to better expose the brain. The organs were removed with forceps and the preparation was immediately used for imaging.

Intracellular Ca$^{2+}$ imaging was performed using a monochromator-based system consisting of a Polychrome V monochromator (TILL Photonics GmbH, Germany), an upright microscope (Olympus, U-TV1X-2, Japan) and a 10X water immersion objective. Images were obtained with an iXon3 888 (Andor, Germany) camera controlled by LiveAcquisition software (TILL Photonics GmbH, Germany). The bath temperature was controlled by a SC-20 dual in-line heater/cooler (Warner Instruments, USA) and an Objective Heater® System (Bioptech, USA). The data were classified semi-automatically using a function programmed in MATLAB (MathWorks, MA) and analyzed with Origin 7.0 (OriginLab Corporation, Northamptom, MA, USA).

### Primary cultureof *Drosophila* larval neurons

Primary neuronal cultures were obtained by dissection of the brain complex of third instar larvae, as described elsewhere (*Harzer et al., 2013*). Dissociated neurons were plated as a 50 µl drop in the center of a coverslip previously coated with a poly-D-lysine/laminin solution. Approximately 40 larvae were dissociated to seed 10 coverslips. Primary neurons were allowed to attach on a coverslip for 2 hr at 25 °C after dissociation. Imaging experiments were performed 3 hr after seeding the cells.

### Immunostaining and intracellular Ca$^{2+}$ imaging in cultured primary neurons

Primary neurons in coverslips were washed three times with PBS and subsequently fixed with 4% paraformaldehyde (PFA) for 20 min. After 10 min incubation with 0.1 M glycine, the cells were permeabilize with 0.1% Triton X-100, followed by 20 min incubation with 3% BSA to reduce unspecific protein bindings. Neuronal subset was stained using a rat anti-Elav (Molecular Probes, 1 hr, dilution 1:100), followed by 30 min incubation with a donkey anti-rat Alexa 488 antibody. Three washes with PBS were used to rinsed cells during all experimental steps described during fixation, permeabilization and staining. After immunostaining coverslips were covered with Vectashield mounting medium (Vector Labs) and imaged on an A1-R confocal (Nikon) mounted on a Ti-2000 inverted microscope (Nikon). The images were processed using ImageJ.

Experiments were performed at 25 °C using a standard Krebs solution containing (in mM): 150 NaCl, 6 KCl, 1 MgCl$_2$, 1.5 CaCl$_2$, 10 glucose, 10 HEPES and titrated to pH 7.4. In experiments on cells expressing GCaMP5 the fluorescence was measured during excitation at 488 nm using a Nikon Eclipse Ti microscope (Nikon) and the NIS Elements 4.30 software. In experiments on cells isolated from wild type flies we used Fura2 as Ca$^{2+}$ indicator. Neurons were functionally identified at the end of each experiment by their responsiveness to the application of an extracellular solution containing high K$^+$ concentration. The data were classified semi-automatically using a function programmed in MATLAB (MathWorks, MA) and analyzed with Origin 7.0 (OriginLab Corporation, Northamptom, MA, USA).

### Immunostainings of fly tissue

For the proboscis staining adult heads were dissected in phosphate buffered saline (PBS) and fixed in 3.7% formaldehyde in PBT (PBS + TritonX100 0.1%) for 15 min. The samples were subsequently rinsed three times in PBT and the proboscis were detached from the heads before being blocked in PAX-DG for 1 hr. Following these steps, the samples were incubated with mouse anti-GFP (Roche) and rabbit anti-dsRED (Clonetech) diluted in PAX-DG overnight at 4 °C. This incubation was followed by three washes with PBT and a subsequent incubation with the appropriate fluorescent secondary antibodies for 2 hr at 25 °C. After three rinses in PBT, the proboscis were transferred in 50% Glycerol diluted in PBS and then mounted in Vectashield (Vector Labs) mounting medium.

For the legs staining the procedure was modified as follows. Flies were fixed in 3.7% formaldehyde in PBT (PBS + TritonX100 3%) for 4 hr, then subsequently rinsed three times in PBT. The legs were then separated from the body and left O/N in the fixing solution.

### Culture and transfection of HEK293T cells

Human embryonic kidney cells, HEK293T, were seeded on 18 mm glass coverslips coated with poly-L-lysine (0.1 mg/ml) and grown in Dulbecco's modified Eagles medium containing 10% (v/v) fetal calf

serum, 2 mM L-glutamine, 2 U/ml penicillin and 2 mg/ml streptomycin at 37 °C in a humidity controlled incubator with 10% $CO_2$. Cells were transiently transfected using Trans-IT-293 reagents (Mirus, Madison, MI, USA) with dTRPA1-A or dTRPA1-B (kindly provided by Paul Garrity) cloned into the pCAGGSM2-IRES-GFP vector.

## Intracellular $Ca^{2+}$ imaging in HEK293T cells

For intracellular $Ca^{2+}$ imaging experiments cells were incubated at 37 °C with 2 µM Fura2-AM ester for 30 min before the recordings. Intracellular $Ca^{2+}$ concentration was measured on an Olympus CellM system at 23 °C. Fluorescence was measured during excitation at 340 and 380 nm, and after correction for the individual background fluorescence signals, the ratio of the fluorescence at both excitation wavelengths (F340/F380) was monitored. In all experiments transfected cells were identified by GFP expression and sensitivity to the TRPA1 agonist AITC.

## Patch-clamp experiments

Whole-cell membrane currents were measured at 23 °C with an EPC-10 patch-clamp amplifier and the softwares Pulse (HEKA, Lambrecht/Pfalz, Germany) and Clampex (Axon Instruments, Sunnyvale CA, US). Currents were digitally filtered at 2.9 kHz, acquired 20 kHz and stored for off-line analysis on a personal computer. Cells were recorded in an extracellular solution containing (in mM): 140 NaCl, 5 KCl, 10 HEPES, 2 $CaCl_2$, 2 $MgCl_2$, 10 glucose, pH titrated to 7.4 with NaOH. The pipette solution contained (in mM): 120 Cs-Aspartate, 5 EGTA, 10 HEPES, 1 $MgCl_2$, pH titrated to 7.4 with CsOH. Non-transfected HEK cells were used as control. Whole-cell currents were elicited using a 200 ms voltage ramp from -110 mV to +110 mV every 2 s from a holding potential of -40 mV. $NMDG^+$ (N-methyl-D-glucamine) was used to monitor the size of the leak currents during the patch-clamp recordings (*Meseguer et al., 2011*). Electrophysiological data were analyzed using WinASCD software (Guy Droogmans, KU Leuven) and Origin (OriginLab Corporation, Northamptom, MA, USA). Origin was also used for statistical analysis and data display.

## Reagents

We used LPS extracted from *E. coli*, strains 055:B5 and 0127:B8. AITC was kept at 4 °C as a 100 mM stock solution in ethanol and fresh dilutions were prepared daily. All chemicals were purchased from Sigma-Aldrich (Bornem, Belgium).

## Acknowledgements

We are very grateful to Paul Garrity for reagents and discussion of the data during the initial part of the project and to Patrik Verstreken for the *NSyB-Gal4;UAS-GCaMP3* flies. We thank Zeynep Okray for her help with larval dissections, Koenraad Philippaert for his help with in vivo $Ca^{2+}$ imaging experiments, and members of the laboratories of Ion Channel Research (KU Leuven) and Neurogenetics (VIB) for helpful discussions. This work was funded by grants from the Belgian Federal Government (Belspo; IUAP P7/13), the Research Council of the KU Leuven (GOA/14/011, OT/12/091 and PF-TRPLe), VIB and the Research Foundation-Flanders (FWO) (G.0702.12, G.0C77.15, G.0680.10, G.0681.10, G.0503.12, G.0654.15, G.0761.10N, G.0596.12 and G.0565.07). AS and YAA hold Postdoctoral Mandates from KU Leuven. BB is funded by a Ph.D. grant of the Agency for Innovation by Science and Technology (IWT, Flanders, Belgium), GL by the FliAct Marie-Curie Initial Training Network, and LF is by a fellowship from the VIB International Ph.D. Program. BAH is an Allen Distinguished Investigator and an Einstein Visiting Fellow of the Berlin Institute of Health.

## Additional information

### Funding

| Funder | Grant reference number | Author |
|---|---|---|
| Vlaams Instituut voor Biotechnologie | | Alessia Soldano<br>Luis Franco<br>Guangda Liu<br>Natalia Mora<br>Emre Yaksi<br>Bassem A Hassan |
| Fonds Wetenschappelijk Onderzoek | G.0702.12 | Alessia Soldano<br>Yeranddy A Alpizar<br>Brett Boonen<br>Alejandro López-Requena<br>Natalia Mora<br>Thomas Voets<br>Rudi Vennekens<br>Bassem A Hassan<br>Karel Talavera |
| Fonds Wetenschappelijk Onderzoek | G.0C77.15 | Alessia Soldano<br>Yeranddy A Alpizar<br>Brett Boonen<br>Alejandro López-Requena<br>Natalia Mora<br>Thomas Voets<br>Rudi Vennekens<br>Bassem A Hassan<br>Karel Talavera |
| Fonds Wetenschappelijk Onderzoek | G.0680.10 | Alessia Soldano<br>Yeranddy A Alpizar<br>Brett Boonen<br>Alejandro López-Requena<br>Natalia Mora<br>Thomas Voets<br>Rudi Vennekens<br>Bassem A Hassan<br>Karel Talavera |
| Fonds Wetenschappelijk Onderzoek | G.0680.10 | Alessia Soldano<br>Yeranddy A Alpizar<br>Brett Boonen<br>Alejandro López-Requena<br>Natalia Mora<br>Thomas Voets<br>Rudi Vennekens<br>Bassem A Hassan<br>Karel Talavera |
| Fonds Wetenschappelijk Onderzoek | G.0680.10 | Alessia Soldano<br>Yeranddy A Alpizar<br>Brett Boonen<br>Alejandro López-Requena<br>Natalia Mora<br>Thomas Voets<br>Rudi Vennekens<br>Bassem A Hassan<br>Karel Talavera |
| Fonds Wetenschappelijk Onderzoek | G.0681.10 | Alessia Soldano<br>Yeranddy A Alpizar<br>Brett Boonen<br>Alejandro López-Requena<br>Natalia Mora<br>Thomas Voets<br>Rudi Vennekens<br>Bassem A Hassan<br>Karel Talavera |

| | | |
|---|---|---|
| Fonds Wetenschappelijk Onderzoek | G.0503.12 | Alessia Soldano<br>Yeranddy A Alpizar<br>Brett Boonen<br>Alejandro López-Requena<br>Natalia Mora<br>Thomas Voets<br>Rudi Vennekens<br>Bassem A Hassan<br>Karel Talavera |
| Fonds Wetenschappelijk Onderzoek | G.0654.15 | Alessia Soldano<br>Yeranddy A Alpizar<br>Brett Boonen<br>Alejandro López-Requena<br>Natalia Mora<br>Thomas Voets<br>Rudi Vennekens<br>Bassem A Hassan<br>Karel Talavera |
| Fonds Wetenschappelijk Onderzoek | G.0761.10N | Alessia Soldano<br>Yeranddy A Alpizar<br>Brett Boonen<br>Alejandro López-Requena<br>Natalia Mora<br>Thomas Voets<br>Rudi Vennekens<br>Bassem A Hassan<br>Karel Talavera |
| Fonds Wetenschappelijk Onderzoek | G.0596.12 | Alessia Soldano<br>Yeranddy A Alpizar<br>Brett Boonen<br>Alejandro López-Requena<br>Natalia Mora<br>Thomas Voets<br>Rudi Vennekens<br>Bassem A Hassan<br>Karel Talavera |
| Fonds Wetenschappelijk Onderzoek | G.0565.07 | Alessia Soldano<br>Yeranddy A Alpizar<br>Brett Boonen<br>Alejandro López-Requena<br>Natalia Mora<br>Thomas Voets<br>Rudi Vennekens<br>Bassem A Hassan<br>Karel Talavera |
| KU Leuven | GOA/14/011 | Alessia Soldano<br>Yeranddy A Alpizar<br>Brett Boonen<br>Luis Franco<br>Alejandro López-Requena<br>Guangda Liu<br>Natalia Mora<br>Emre Yaksi<br>Thomas Voets<br>Rudi Vennekens<br>Bassem A Hassan<br>Karel Talavera |
| European Commission | IUAP P7/13 | Alessia Soldano<br>Yeranddy A Alpizar<br>Brett Boonen<br>Luis Franco<br>Alejandro López-Requena<br>Guangda Liu<br>Natalia Mora<br>Emre Yaksi<br>Thomas Voets<br>Rudi Vennekens |

| KU Leuven | OT/12/091 | Alessia Soldano |
| | | Yeranddy A Alpizar |
| | | Brett Boonen |
| | | Luis Franco |
| | | Alejandro López-Requena |
| | | Guangda Liu |
| | | Natalia Mora |
| | | Emre Yaksi |
| | | Thomas Voets |
| | | Rudi Vennekens |
| | | Bassem A Hassan |
| | | Karel Talavera |
| KU Leuven | PF-TRPLe | Alessia Soldano |
| | | Yeranddy A Alpizar |
| | | Brett Boonen |
| | | Luis Franco |
| | | Alejandro López-Requena |
| | | Guangda Liu |
| | | Natalia Mora |
| | | Emre Yaksi |
| | | Thomas Voets |
| | | Rudi Vennekens |
| | | Bassem A Hassan |
| | | Karel Talavera |

The funders had no role in study design, data collection and interpretation, or the decision to submit the work for publication.

### Author contributions

AS, YAA, Conception and design, Acquisition of data, Analysis and interpretation of data, Drafting or revising the article; BB, LF, GL, NM, Acquisition of data, Analysis and interpretation of data; AL-R, Drafting or revising the article, Contributed unpublished essential data or reagents; EY, Conception and design, Analysis and interpretation of data; TV, RV, Analysis and interpretation of data, Drafting or revising the article; BAH, KT, Conception and design, Analysis and interpretation of data, Drafting or revising the article

### Author ORCIDs

Alejandro López-Requena, http://orcid.org/0000-0002-6951-8321
Thomas Voets, http://orcid.org/0000-0001-5526-5821
Bassem A Hassan, http://orcid.org/0000-0001-9533-4908

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
