## [Decision Letter]

Thank you for submitting your work entitled "Gustatory-mediated avoidance of bacterial lipopolysaccharides via TRPA1 activation in *Drosophila*" for consideration by *eLife*. Your article has been favorably evaluated by K VijayRaghavan (Senior editor) and three reviewers, one of whom is a member of our Board of Reviewing Editors. One of the three reviewers has agreed to reveal his identity: Hubert Amrein.

The reviewers have discussed the reviews with one another and the Reviewing Editor has drafted this decision to help you prepare a revised submission.

Summary:

This manuscript examines mechanisms for detection of a bacterial lipid, LPS. The authors show that flies avoid LPS, this avoidance requires bitter gustatory sensory neurons, and TRPA1 expressed in bitter neurons is essential for aversion. By misexpression studies, the authors show that TRPA1 is activated by LPS. The study is interesting, identifying a potential receptor and sensory mechanism in *Drosophila* for bacterial detection (with evident selective advantage for an animal that feeds upon microbe-rich substrates), and broadening the sensory profile of the well-characterized Gr66a gustatory "aversion" neurons.

Overall, the behavioral experiments are rigorous and convincing. The functional studies of LPS-mediated activation of TRPA1 require additional experiments to link the behavioral and cellular phenotypes.

Essential revisions:

1) Straight-forward methods (Ca^2+^ imaging and electrophysiological recordings) are established to monitor cellular responses of taste neurons. The authors need to monitor the responses of bitter gustatory neurons in wt, *trpA1* mutant, and *trpA1^-/-^; Gr66a-GAL4, UAS-trpA1* rescue flies to show that LPS activates these neurons and requires TRPA1. This is essential to link sensory detection of LPS to the behavioral response.

2) Figure 3 legend states that the experiment was done in isolated neurons expressing GCaMP and RFP in *dTRPA1-Gal4* neurons. In the text, it states that primary cultures were examined using Fura2 (Results and Discussion, fifth paragraph). The experiments should be done using GCaMP expressed selectively in *dTRPA1-Gal4* cells (if they were not). The number of cells responding in *trpa1* mutant and WT with HC is confusing (Figure 3) and will perhaps be resolved if only TRPA1-expressing cells are being studied.

3) LPS are very different in structure and much larger than AITC and other TrpA1 ligands. Discussion of the structure of LPS and discussion of which part of LPS (polyglycan, lipid) is critical to activate TrpA1 would be useful. In addition, LPS concentration used (0.1%) should be compared to the concentration of LPS in bacterially contaminated food.

[Editors' note: further revisions were requested prior to acceptance, as described below.]

Thank you for resubmitting your work entitled "Gustatory-mediated avoidance of bacterial lipopolysaccharides via TRPA1 activation in *Drosophila*" for further consideration at *eLife*. Your revised article has been favorably evaluated by K VijayRaghavan as the Senior editor and a Reviewing editor.

The manuscript has been improved but there is one remaining issue that needs to be addressed before acceptance, as outlined below:

The calcium imaging data added in the revision should be removed. The responses are too inconsistent to interpret and the manuscript is stronger without this data. If the resubmission were modified to state that the experiment was attempted but that there were challenges associated with stimulating mouthpart neurons and direct brain application gave varying results, this would enhance the manuscript.

---

## [Author Response]

Essential revisions:

1) Straight-forward methods (Ca^2+^ imaging and electrophysiological recordings) are established to monitor cellular responses of taste neurons. The authors need to monitor the responses of bitter gustatory neurons in wt, trpA1 mutant, and trpA1^-/-^; Gr66a-GAL4, UAS-trpA1 rescue flies to show that LPS activates these neurons and requires TRPA1. This is essential to link sensory detection of LPS to the behavioral response.

As previously described in the literature (Kang et al., Nature 2010, 464:597-600) and as demonstrated in the newly added data (new Figure 3 and Figure 3—figure supplement 1, the immunofluorescence from labellum and leg), *dTrpA1* is expressed in a subset of *Gr66a* bitter-sensing gustatory neurons present in the labellum and in the pharynx, but not in the legs. We are persuaded that the neurons involved in the LPS detection are the ones present in the pharynx, where they detect potentially harmful substances upon food ingestion, before reaching the digestive tract. This is strongly supported by the fact that LPS is avoided by flies when examined with a two-choice assay, but does *not* inhibit the proboscis extension reflex (new Figure 3—figure supplement 2). Together, these data strongly support the involvement of internal sensory neurons in the detection of LPS rather than labial sensory neurons. It was not possible to perform electrophysiological recordings because this technique allows only recording from external sensory neurons that are easily accessible (Benton and Dahanukar Cold Spring Harbor protocols 2011, 2011:839-850). To our knowledge there reports of recordings performed from internal sensory neurons. Similarly, we did not find any example in the literature of Ca^2+^ imaging used to monitor the responses of pharyngeal GRNs. However, performing intracellular Ca^2+^ imaging in a perfusion system allowed us to analyze the responses of the neuronal terminals in the sub-esophageal ganglion (SOG) upon stimulation of the gustatory nerves and the neuronal terminals themselves, therefore overcoming the accessibility issue.

We focused on using Ca^2+^ imaging to determine if LPS elicits Ca^2+^ responses to LPS in the SOG. We visualized the projections of bitter-sensing neurons in the SOG by opening a window on the fly head cuticles and submerging the fly head into AHL solution (Marella et al., Neuron 2006, 49:285-295). We analyzed *Gr66aGal4:UAS-GCaMP6m* flies and used caffeine as a positive control for bitter-sensing gustatory neuron activation (Marella et al., Neuron 2006, 49:285-295) and allyl isothiocyanate (AITC) as a control for dTRPA1 activation. As shown in the new Figure 3, we found that application of LPS induces neuronal activation. Not surprisingly, the responses were variable in amplitude and duration, and we could not observe them in all analyzed brains. We are convinced that this was not due to technical issues because all samples responded robustly and consistently to caffeine and AITC. We believe that the variability of the responses to LPS is due to the intrinsic nature of the molecule itself. LPS, is a very large molecule consisting of a lipophilic moiety (lipid A), a polysaccharide core, and a highly variable O-polysaccharide. Therefore, LPS may not diffuse well through the tissue, in contrast to classic tastants such as caffeine. In the experiments we performed LPS was applied to an in vivo preparation where it has to access the projection of bitter-sensing neurons that are located in a deep part of the preparation. Moreover, as shown in the Ca^2+^ imaging experiment in Figure 3, LPS induces a specific but rather mild activation of *dTrpA1*-expressing neurons.

In conclusion, our evidence strongly supports the specificity of the activation of Gr66a neurons by LPS, but due to the variability of the effects observed we do not think that analyzing this response in the *dTrpA1* null animals and genetic rescue would be much more informative. Moreover, our other data clearly demonstrate that *dTrpA1* is necessary and sufficient for LPS detection as shown in Ca^2+^ imaging experiments (Figure 3) and specifically in the Gr66a neurons in the two-choice assay (Figure 1).

2) Figure 3 legend states that the experiment was done in isolated neurons expressing GCaMP and RFP in dTRPA1-Gal4 neurons. In the text, it states that primary cultures were examined using Fura2 (Results and Discussion, fifth paragraph). The experiments should be done using GCaMP expressed selectively in dTRPA1-Gal4 cells (if they were not).

There was indeed a mistake in the text, thank you for pointing this out. In these experiments we used neurons isolated from animals expressing RFP and GCaMP5 under the control of *TrpA1-Gal4*. The text referring to this figure is now corrected (Figure 4 in the current version of the manuscript).

The number of cells responding in trpa1 mutant and WT with HC is confusing (Figure 3) and will perhaps be resolved if only TRPA1-expressing cells are being studied.

It is indeed true that the responses to LPS and NEM are not completely absent in the neurons isolated from *TrpA1* mutant flies, nor in wild type neurons exposed to the TRPA1 inhibitor. By no means have we wanted to claim that TRPA1 is the only target for LPS and NEM in these neuronal cultures. Please, note that this preparation serves as a model for endogenous TRPA1 expression, but may bare no direct relevance to the avoidance to LPS observed in the behavioral experiments.

However, these experiments clearly show that the neuronal population responding to both LPS and NEM is reduced by genetic or pharmacological ablation of TRPA1. To confirm the role of TRPA1 in the responses to LPS we performed another series of experiments, using the ratiometric dye Fura2, and AITC instead of NEM as reference dTRPA1 agonist. The results, now shown in the new Figure 4—figure supplement 3 support the idea that TRPA1 contributes to the responses to LPS in a native expression system. Activation of TRPA1 by LPS was clearly demonstrated by the results of our patch-clamp experiments, which show that LPS enhances dTRPA1 currents in a heterologous expression system.

3) LPS are very different in structure and much larger than AITC and other TrpA1 ligands. Discussion of the structure of LPS and discussion of which part of LPS (polyglycan, lipid) is critical to activate TrpA1 would be useful.

As suggested by the reviewers, we now add a brief comment about this issue: “Previous results suggest that LPS activates mammalian TRPA1 channels by inducing mechanical perturbations in the plasma membrane upon insertion of the lipophilic moiety of the molecule [Meseguer et al., 2014]. It is conceivable that LPS activates dTRPA1 channels via the same mechanism, but additional experiments are required to verify this.”

We would like to point out that, given the unusual chemical structure of LPS, the elucidation of the molecular mechanism underlying its action on TRPA1 represents a major undertaking, requiring numerous and sophisticated biophysical approaches that go far beyond the scope of the present study.

In addition, LPS concentration used (0.1%) should be compared to the concentration of LPS in bacterially contaminated food.

LPS concentrations can be expected to be very high, especially in the ecological niches of fruit flies. Although we did not attempt to measure these concentrations ourselves, the literature reports values between 250 and 800 µg/ml in water used to rinse fruits and vegetables contaminated with *E. coli* (Wang et al., J Microbial Biochem Technol 2011, 3:26-29). It is therefore expected that the concentrations in the surface of the food would be much higher. Considering this, we are confident that the concentrations we used in our behavioral experiments, and in our experiments in vitro are relevant for real scenarios. We now include a statement on this in the Materials and methods section (Binary choice food preference) to comply with the suggestion of the referee.

Furthermore, in our revised manuscript we now include new results of egg laying experiments aimed at testing the ability of flies to avoid bacteria (*E. coli*). As shown in Figure 2, control animals show a marked preference for laying eggs on food without bacteria. In contrast, *dTrpA1* null mutants showed no bacterial avoidance, suggesting that dTRPA1 is important for bacterial detection. Taken together, these arguments indicate that the concentration of LPS we used can be detected by fly as sign of bacterial infection and that the LPS detection is a physiologically relevant mechanism that allows avoidance of infected food.

[Editors' note: further revisions were requested prior to acceptance, as described below.]

The manuscript has been improved but there is one remaining issue that need to be addressed before acceptance, as outlined below:

The calcium imaging data added in the revision should be removed. The responses are too inconsistent to interpret and the manuscript is stronger without this data. If the resubmission were modified to state that the experiment was attempted but that there were challenges associated with stimulating mouthpart neurons and direct brain application gave varying results, this would enhance the manuscript.

We have revised the manuscript based on the comments of Reviewing Editor. We removed the calcium imaging data from the text and figures and added the following sentence:

“We attempted to record neuronal responses in these neurons using flies expressing the genetically encoded Ca^2+^ indicator GCaMP6m in *Gr66a* neurons but Ca^2+^ imaging access to these neurons proved impossible in our assays. Next, we attempted direct brain stimulation. All preparations (5/5) responded robustly to the application of the dTRPA1 agonist allyl isothiocyanate (AITC) or the classical bitter compound caffeine, indicating that they were healthy. However, application of LPS gave varying results (small responses in 40% (2/5) of the flies; data not shown) precluding definitive conclusions.”